# Co-Targeting PD-1 and IL-33/ST2 Pathways for Enhanced Acquired Anti-Tumor Immunity in Breast Cancer

**DOI:** 10.3390/ijms26199600

**Published:** 2025-10-01

**Authors:** Marina Z. Jovanović, Milena Jurišević, Milan Jovanović, Nevena Gajović, Miodrag Jocić, Marina M. Jovanović, Boško Milev, Krstina Doklestić Vasiljev, Ivan Jovanović

**Affiliations:** 1Center for Molecular Medicine and Stem Cell Research, Faculty of Medical Sciences, University of Kragujevac, 34000 Kragujevac, Serbia; marina.z.jovanovic@fmn.kg.ac.rs (M.Z.J.); ivanjovanovic77@gmail.com (I.J.); 2Department of Otorhinolaryngology, Faculty of Medical Sciences, University of Kragujevac, 34000 Kragujevac, Serbia; 3Department of Pharmacy, Faculty of Medical Sciences, University of Kragujevac, 34000 Kragujevac, Serbia; 4Department of Abdominal Surgery, Military Medical Academy, 11000 Belgrade, Serbia; milan.jovanovic@vma.mod.gov.rs (M.J.); milevbosko@gmail.com (B.M.); 5Medical Faculty of the Military Medical Academy, University of Defense, 11000 Belgrade, Serbia; jocicmiodrag@gmail.com; 6Institute for Transfusiology and Haemobiology, Military Medical Academy, 11000 Belgrade, Serbia; 7Department of Internal Medicine, Faculty of Medical Sciences, University of Kragujevac, 34000 Kragujevac, Serbia; marinna034@gmail.com; 8Faculty of Medicine, University of Belgrade, 11000 Belgrade, Serbia; krstinadoklestic@gmail.com; 9Clinic for Emergency Surgery, University Clinical Center of Serbia, 11000 Belgrade, Serbia; 10Faculty of Medicine, University of East Sarajevo, 73300 Foca, Bosnia and Herzegovina

**Keywords:** T cells, macrophages, breast cancer, IL-33, anti-PD-1

## Abstract

Despite advances in immunotherapy, the treatment of breast cancer still remains a major global problem. In a previous study, we showed that co-blockade of Interleukin-33/ST2 and Programmed death-1/Programmed death-ligand (PD-1/PD-L) signaling pathways strongly slows progression by enhancing the antitumor capacity of natural killer (NK) cells. The main aim of this study is to elucidate the exact effect of co-blockade on the T lymphocyte and macrophage effector cells. 4T1 cells were used to induct breast cancer in female BALB/C and BALB/C ST2^−/−^ mice. The mice, both BALB/C and BALB/C ST2^−/−^, were treated with anti-PD-1 antibody on certain days. After the mice were sacrificed, T cells and macrophages were analyzed using flow cytometry; dual co-blockade increased significantly the percentage of M1 macrophages in the tumor microenvironment, followed by an increase in expression of CD86^+^ and TNFα^+^. T cell accumulation was significantly higher in the spleen and within the tumor microenvironment, with elevation in activation markers such as Interleukin-17, CD69, NKG2D, and FasL and a decrease in Interleukin-10 and FoxP3 expression. Co-blockade of the PD-1/PD-L axes and IL-33/ST2 axes shows promising results in reestablishing an effective immune response and offers a new perspective on improving immune response to breast carcinoma.

## 1. Introduction

Breast cancer is the most common type of cancer among women, as well as the most common cause of death, despite advances in prevention and treatment modalities. It persists as a major health problem in women aged 35 to 54 years [1]. Generally speaking, there has been a marked improvement in treating cancers via the introduction of cancer immunotherapy [2]. Targeted breast cancer immunotherapy has been a challenge, since breast cancer often presents with vast heterogeneity [3]. On the other hand, utilizing one’s immune response and stimulating overall anti-tumor immune response seems to be a promising approach to malignancy treatment [4].

Nevertheless, cancer immunotherapy has been evolving rapidly and is certainly gaining more attention in the clinical setting [5]. Immune checkpoint blockade, using anti-CTLA-4 (Cytotoxic T-lymphocyte-associated protein 4) and especially anti-programmed death-1 (PD-1) therapy, has shown efficacy in cancer treatment [6]. The anti-PD-1 blockade was approved for treating triple-negative breast cancer in 2021 [7]. The main advantage, but also the disadvantage, is that blockade of the PD-1/PD-L (programmed death-ligand) axis is immunomodulatory. Although it amplifies anti-tumor immune response, it can also induce extensive immune-related adverse events (IrAEs) [8]. IrAEs, such as uveitis, enterocolitis, and thyroid abnormalities, including hyper/hypothyroidism, thyrotoxicosis, and even so called “thyroid storm”, can be a challenge to treat in a cancer patient [9]. The fine-tuning between achieving desired immunomodulatory, anti-tumor effects and avoiding IrAEs is still a matter of debate [10].

Triggering one’s anti-tumor immune response and modulating production of the cytokines involved in the pathogenesis of breast cancer might yield a more effective treatment [11]. One of the more recently acknowledged cytokines that exerts tremendous impact on breast cancer is interleukin 33 (IL-33) [12]. IL-33 has been widely addressed in the context of various cancers [13]. Primarily identified as pro-inflammatory cytokine, IL-33 has been reported to be elevated in the sera of breast cancer patients compared to healthy controls, potentially serving as a prognostic marker [13,14]. In addition, it has been observed that IL-33 signaling can promote breast carcinoma progression by inducing metastasis progression and neoangiogenesis [14,15]. Also, it has been shown that overstimulation of the IL-33/ST2 axis strongly correlates with the accumulation of immunosuppressive cells within tumor tissue, thus creating an environment ideal for breast carcinoma expansion [16]. We have shown previously that deletion of IL-33/ST2 signaling enhances innate anti-tumor immunity and subsequently decelerates tumor growth and progression [17]. Having in mind that separate blockade both of these axes has its advantages in stimulating anti-tumor immune responses via different pathways, we investigated the effect of combined blockade on the overall anti-tumor response. Since PD-1/PD-L interactions are crucial for suppressing T-cell-mediated antitumor immunity, and the IL-33/ST2 signaling pathway can promote cancer progression by causing inflammation and evading the immune system, we tested how simultaneous blockade of these two axes could modulate immune cell phenotypes and functional responses within the spleen and tumor microenvironment.

In our previous study, we demonstrated that co-blockade of these signaling pathways significantly slowed breast and colon cancer progression by enhancing the antitumor capacity of innate cells, mainly by activating natural killer (NK) cells directly and by overall lowered immunosuppression [18]. The upregulation of NF-κB (Nuclear Factor kappa-light-chain-enhancer of activated B cells) and STAT3 (Signal Transducer and Activator of Transcription 3) signaling pathways in NK cells, which in turn stimulates the expression of proinflammatory and cytotoxic molecules like FasL, NKG2D, NKp46, perforin, Interleukin-17, and IFN-γ (Interferon gamma), is at least partially responsible for this effect. The main aim of the present study is to clarify the effects of dual blockade of the PD-1/PD-L and IL-33/ST2 on innate and acquired anti-tumor immunity effector cells, T cells, and macrophages, in a murine breast cancer model.

## 2. Results

### 2.1. Anti-PD-1 Therapy Enhances the Accumulation and Polarization Toward M1 Macrophages in the Tumor Microenvironment of ST2^−/−^ Mice

Firstly, we analyzed the functional phenotype of spleen macrophages. The percentage of F4/80^+^ cells did not differ significantly between groups (Figure 1A; the gating strategy used to identify F4/80^+^ macrophages is provided in Appendix A). The expression of TNFα in these cells did not differ between groups (Figure 1B).

In the tumor microenvironment, the percentage of F4/80^+^ cells was significantly higher in ST2^−/−^ anti-PD-1-treated mice in comparison to all other groups (*p* < 0.05; Figure 2A; Appendix A). The percentage of CD86^+^ and TNFα^+^ F4/80^+^ cells was the highest in ST2^−/−^ anti-PD-1-treated mice (*p* < 0.05; Figure 2B,C).

### 2.2. Anti-PD-1 Therapy Increases the Accumulation of T Cells and Expression of Activation Molecules in the Spleen of ST2^−/−^ Mice

We further analyzed T cells in spleens and primary tumors, after 4T1 tumor cell application. The percentage of CD3^+^CD49b^−^ T cells in the spleens of tumor-bearing mice was the highest in the ST2^−/−^ anti-PD-1-treated group (*p* < 0.05; Figure 3A; the gating strategy used to identify CD3^+^CD49b^−^ T cells is provided in Appendix A). The percentage of NKG2D^+^, CD107a^+^, and IL-17^+^ CD3^+^CD49b^−^ T cells was the highest in the spleens of ST2^−/−^ anti-PD-1-treated mice (*p* < 0.05; Figure 3B–D).

### 2.3. Anti-PD-1 Therapy Alters the Phenotype of T Cells in the Tumor Microenvironment of ST2^−/−^ Mice

Additionally, functional phenotypes of T cells were analyzed in the tumor microenvironment. There was no difference in the accumulation of CD3^+^CD49b^−^ T cells in the tumor microenvironment among groups (Figure 4A; Appendix A). The percentage of IL-10^+^ and Foxp3^+^ CD3^+^CD49b^−^ T cells was the lowest in the primary tumor of ST2^−/−^ anti-PD-1-treated mice compared to untreated groups (*p* < 0.05) (Figure 4B,C). In addition, the percentages of CD69^+^, NKG2D^+^, and FasL^+^ CD3^+^CD49b^−^ T cells within primary tumors were highest in ST2^−/−^ anti-PD-1-treated mice compared to other groups (*p* < 0.05) (Figure 4D–F). Although a slight increase in the percentages of CD69^+^, NKG2D^+^, and FasL^+^ CD3^+^CD49b^−^ T lymphocytes in the tumor microenvironment of WT mice after anti-PD-1 treatment were observed compared to untreated WT mice, the difference did not reach statistical significance.

## 3. Discussion

Breast cancer is one of most commonly diagnosed cancers worldwide [1]. It is still one of the most challenging malignancies to treat due to various mechanisms of cancer spreading and progression [19]. Chemoresistance in breast cancer is a well recognized and increasingly observed phenomenon in clinical practice, making disease control and treatment more challenging [20]. Many authors agree that chemoresistance is a result of insufficient anti-tumor immune response [21,22,23]. Anti-PD-1 therapy has been officially approved for breast cancer as an induction therapy for early-stage, high-risk triple negative breast cancer before surgery, and for recurrent, metastatic, or unresectable breast cancer. On the other hand, IL-33/ST2 signaling has been shown to induce cancer progression and metastasis; emerging preclinical evidence suggests that blocking IL-33 may be effective in several cancers, including breast cancer [24,25].

Our previous study illustrated the effect of simultaneous blockade of the PD-1/PD-L and IL-33/ST2 axes on innate anti-tumor immunity. We showed a profound effect on NK cells that were directly upregulated by the co-blockade, alongside decreased activation of immunosuppressive cells, which ultimately led to slower breast cancer growth [18].

In our present study, we firstly analyzed the macrophages in the spleen and the tumor microenvironment (TME). The dual blockade did not affect the accumulation of macrophages in the spleen, as well as the production of TNFα (Figure 1). Further, there was a significant increase in percentage of F4/80^+^ macrophages in the dual blockade group within the TME in comparison to all other groups (Figure 2A). In addition, there was elevation in activating TNFα, as well as CD86 within the TME (Figure 2B). Tumor-associated macrophages (TAMs) of breast cancer can either be pro-tumorigenic or anti-tumorigenic, suggesting their important role in breast cancer immunity. It has been suggested that pro-tumorigenic macrophages, classically referred to as M2 macrophages, can induce immunosuppression via the increased production of IL-3, IL-4, IL-10, and TGF-β within the TME [26]. Also, it has been shown that an increased M2 population within the TME leads to poor survival [27]. On the other hand, anti-tumorigenic macrophages, or M1 macrophages, can act in an anti-tumorigenic manner via the expression of co-stimulatory molecules and increased antigen presentation to effector cells, such as T cells [28]. Traditionally, M1 phenotypes of classically activated macrophages are characterized by increased expression of HLA-DR, CD86, and CD68 [29]. It has been shown that elevation in CD86 can increase Th1 activation, thus promoting anti-tumor immune response [30]. Our results indicate that dual blockade tends to polarize TAMs towards M1-like phenotypes, which promotes the reestablishment of a more effective anti-tumor immune response.

When it comes to the mainstay effector cells, T cells are certainly included. It has been postulated that both the axes IL-33/ST2 and PD-1/PD-L can impact the effectiveness of these cells. It has been shown that IL-33 can promote breast cancer metastasis by activating fibroblasts and polarizing T cells towards the pro-tumorigenic T2 phenotype [14]. Our study group has also shown that IL-33 signaling induces breast cancer progression by modulating the TME [16]. On the other hand, various studies have shown that PD-L expression, especially PD-L1 expression, is elevated in many cell types within the TME, including tumor cells and antigen presenting cells [31,32,33]. In terms of dual blockade of these axes, the percentage of splenic T cells is significantly increased (Figure 3A). Also, dual blockade significantly increases the expression of activating molecule NKG2D in splenic T cells (Figure 3B). NKG2D in T cells is thought to enhance TCR signaling, similarly to CD28, and also participates in the formation of memory T cells, therefore contributing to a more effective T cell response [34]. Co-blockade of IL-33/ST2 and PD-1/PD-L also increases the expression of CD107a (Figure 3C), a degranulation marker, which signals the increased cytotoxic activity of T cells [35]. In addition, co-blockade significantly increased the expression of IL-17 in splenic T cells, indicating a potential enhancement of Th17 response (Figure 3C). The dual role of IL-17, as well as Th17 cells, is well known [36]. IL-17 can potentially activate cytotoxic CD8^+^ T cells, employ the innate immune cells, or even convert Th17 cells to the Th1 phenotype, consequently triggering an antitumor response [36,37]. This finding suggests that dual blockade of the PD-1/PD-L and IL-33/ST2 axes may affect T cells’ effector functions and possibly assist in antitumor immunity. In addition, there is emerging data indicating that increased levels of IL-17 during inflammation can trigger monocyte chemoattractant protein–1 (MCP-1) and polarize monocytes to M1 macrophages [38]. We assume that the predominance of the Th17 immune response may facilitate monocyte recruitment into the TME and polarization in the M1 phenotype. Subsequently, M1 monocytes/macrophages may facilitate the induction of T lymphocytes, which closes the positive feedback loop.

When it comes to the tumor microenvironment, dual blockade did not alter the percentage of tumor-infiltrating T cells, but significantly altered their phenotypes (Figure 4). Co-blockade of the IL-33/ST2 and PD-1/PD-L axes significantly decreased the expression of immunosuppressive markers IL-10 and FoxP3 (Figure 4B,C). There are various studies that illustrate the role of IL-10 in cancer progression [39,40,41]. Within the TME, IL-10 exhibits a predominantly immunosuppressive role, and is therefore considered a possible target for cancer immunotherapy [42]. IL-10 facilitates immune escape by suppressing effector T cell function, promoting Treg development, and supporting M2 TAM polarization [39,40,41,42]. Under specific conditions, IL-10 may contribute to potential antitumor effects by enhancing CD8^+^ T cell function [43]. On the other hand, FoxP3 is a well known marker of regulatory T cells that are able to stimulate immunosuppressive mechanisms in the tumor microenvironment [44]. Increased expression of Foxp3 and IL-10 is linked to an immunosuppressive environment that facilitates tumor immune escape [45]. Reduced expression of Foxp3 and IL-10 in induced Tregs after anti-PD-1 treatment has been reported [46]. Dual blockade of IL-33/ST2 and PD-1/PD-L, as we previously reported, resulted in the reduced presence of IL-10⁺ and FoxP3⁺ CD3^−^CD49⁺ NK cells in the spleens and tumor environment [18]. In addition, co-blockade of IL-33/ST2 and PD-1/PD-L resulted in the lowest accumulation of Tregs, including IL-10-producing Tregs, in both the tumor microenvironment and spleen [18]. This change happened alongside the predominance of M1-like TAMs, and likely resulted in enhancing the antitumor immune response. Namely, co-blockade of the IL-33/ST2 and PD-1/PD-L axes increased the expression of NKG2D and CD69 markers, suggesting there is a higher activation rate of T cells within the TME in terms of dual blockade (Figure 4D,E). Also, there is an increase in FasL expression in T cells (Figure 4F). It is known that higher expression of FasL correlates with higher apoptosis of tumor cells, and even increased survival of cancer patients [47,48]. Our results reveal that single blockade affects the change in the functional phenotype of macrophages and T lymphocytes, but this change does not reach statistical significance. Only dual blockade completely induces the phenotypic change in these cells of interest.

While dual blockade was previously shown to boost NK cell-mediated anti-tumor immunity [18], our new results reveal for the first time that dual blockade also changes the T-cell phenotype in the spleen and tumor, and induces polarization of macrophages towards M1-like phenotypes, as illustrated in Figure 5.

It should be noted that the dual blockade of PD-1/PD-L and IL-33/ST2 delays tumor appearance and growth, alongside the direct effect of dual blockade on NK cells we previously reported [18]. In addition, we extended these findings by focusing on acquired antitumor immunity. Based on the illustrated beneficial effects of dual blockade on T cells, as well as M1 macrophages, we suggested a major impact of dual blockade on both innate and acquired anti-tumor immunity. Combined blockade of IL-33/ST2 and PD-1/PD-Lsignificantly alters the tumor microenvironment and stimulates overall anti-tumor immune response. These findings suggest a novel approach to breast cancer therapy, whether it be as adjuvant treatment, possibly overcoming chemoresistance, or as an innovative perspective in breast cancer immunotherapy.

## 4. Materials and Methods

### 4.1. Mice

We used female, BALB/C wild-type (WT) and BALB/C ST2 knockout (ST2^−/−^) mice, 6–8 weeks old. BALB/c ST2 knockout mice were used as endogenous way of blocking IL-33/ST2 axis. Experiments were performed in the Center for Molecular Medicine and Stem Cell Research of Faculty of Medical Sciences, University of Kragujevac, Serbia. The mice were kept under standard laboratory conditions (12-h light-dark cycle, 22 ± 2 °C, and relative humidity 51 ± 5%) throughout the duration of the experiment. The Animal Ethics Board of the Faculty of Medical Sciences, University of Kragujevac, Serbia (N0 01-12336/2; date: 26 October 2018) approved all experiments. All experiments were performed in accordance with the ARRIVE guidelines and EU Directive 2010/63/EU for animal experiments. After tumor inoculation, mice were divided into four experimental groups: untreated wild-type BALB/C mice (WT) (control group) (1), untreated BALB/C ST2 knockout (ST2^−/−^) (endogenous IL-33/ST2 blockade group) (2), wild-type BALB/C anti-PD-1-treated (WT anti-PD-1-treated) (3), and BALB/C ST2 knockout (ST2^−/−^) anti-PD-1-treated mice (ST2^−/−^ anti-PD-1-treated) (4).

### 4.2. 4T1 Tumor Induction

The 4T1 cell line of murine mammary carcinoma was purchased from the American Type Culture Collection (ATCC, Manassas, Virginia, United States). 4T1 cells were cultured in Dulbecco’s Modified Eagle’s Medium (DMEM) (supplemented with 2 mmol/L L-glutamine, 10% heat-inactivated fetal bovine serum (FBS), 1 mmol/L mixed nonessential amino acids (Sigma-Aldrich, Burlington, Massachusetts, United States) and 1 mmol/L penicillin–streptomycin). Before in vivo experiments, cultured 4T1 cells were harvested and washed three times in serum-free PBS (Phosphate-Buffered Saline). 4T1 cells (5 × 10^3^) were inoculated into the 4th mammary fat pad of each mouse, as previously reported [16,18].

#### Anti-PD-1 Antibody Administration

Murine anti-PD-1 antibody (purchased from BioXcell, Lebanon, New Hampshire, United States) was administered intraperitoneally to WT BALB/C and ST2 knockout mice (ST2^−/−^) in total four doses (each dose per mice 150 μg anti-PD-1 antibody dissolved in 150 μL of PBS [18]), while control untreated WT and untreated ST2^−/−^ mice received PBS only (150 μL of PBS). Mice underwent appropriated treatment on the third, sixth, ninth, and eleventh day after tumor inoculation (day 0), as previously reported [18,49].

### 4.3. Flow Cytometric Analyses

As previously described, single-cell suspensions of primary breast tumors and spleens were obtained via enzymatic digestion and by mechanical dispersion, respectively [18,50]. Anti-mouse mAbs specific for CD86 (FUN-1), CD49 (9F10), CD3 (PC3 188A), FasL (MFL3), CD69 (FN50), NKG2D (14-5878-82), and F4/80 (11-4801-85), or isotype-matched controls (BD Pharmingen, NJ/Invitrogen, Carlsbad, CA, USA), were used [18,50]. Cells were also stimulated with ionomycin (500 ng/mL, Sigma-Aldrich, Burlington, MA, USA), phorbol 12-myristate 13-acetate (50 ng/mL, Sigma-Aldrich, Burlington, MA, USA), and GolgyStop (BD Pharmingen, NJ, USA/BD Biosciences, San Jose, CA, USA) for 4 h and stained with fluorochrome-labeled anti-mouse mAbs specific for CD107a (1D4B), Foxp3 (MF23), IL-10 (JES5-16E3), IL-17 (C15.6), and TNFα (MP6-XT22) (BD Pharmingen, NJ, USA/BioLegend San Diego, CA, USA/eBiosciences, San Jose, CA, USA) [18,50]. A FACSCalibur Flow Cytometer (BD Biosciences, San Jose, CA, USA) was used and the data were analyzed using FlowJo (version 10.7.2.; Tree Star) [50]. The complete gating strategy is shown in Appendix A.

### 4.4. Statistical Analysis

Commercially available SPSS (version 23.0) software was used in order to analyze the data. Student’s *t* tests, Mann-Whitney U tests, ANOVA, or Kruskal–Wallis tests were used where appropriate. Data are presented as average ± SEM. Statistical significance was set at *p* < 0.05.

## Figures and Tables

**Figure 1 ijms-26-09600-f001:**
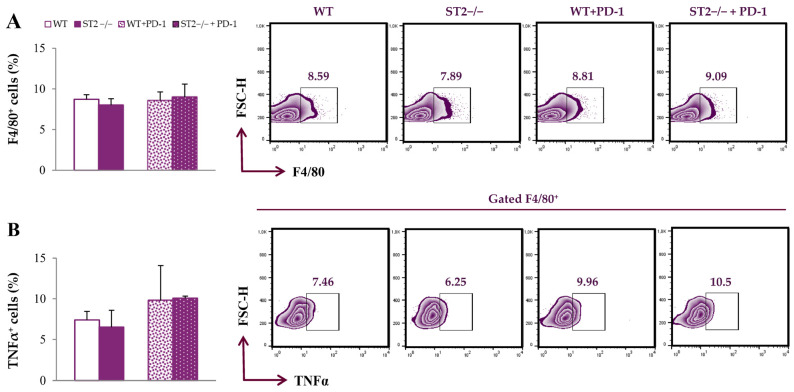
Dual blockade of the PD-1/PD-L and IL-33/ST2 axes does not alter the percentage of spleen-derived macrophages. The graphs and representative plots representing the percentage of F4/80^+^ cells (**A**) and expression of TNFα in F4/80^+^ cells (**B**) derived from the spleens of WT (BALB/C), ST2^−/−^ (BALB/C ST2 knockout), anti-PD-1-treated WT (WT + PD-1) and anti-PD-1-treated ST2^−/−^ (ST2^−/−^ + PD-1) mice. Data are presented as average ± SEM (*n* = 7 mice per group). Statistical significance was tested via the Kruskal–Wallis test and a post hoc Mann–Whitney Rank Sum test.

**Figure 2 ijms-26-09600-f002:**
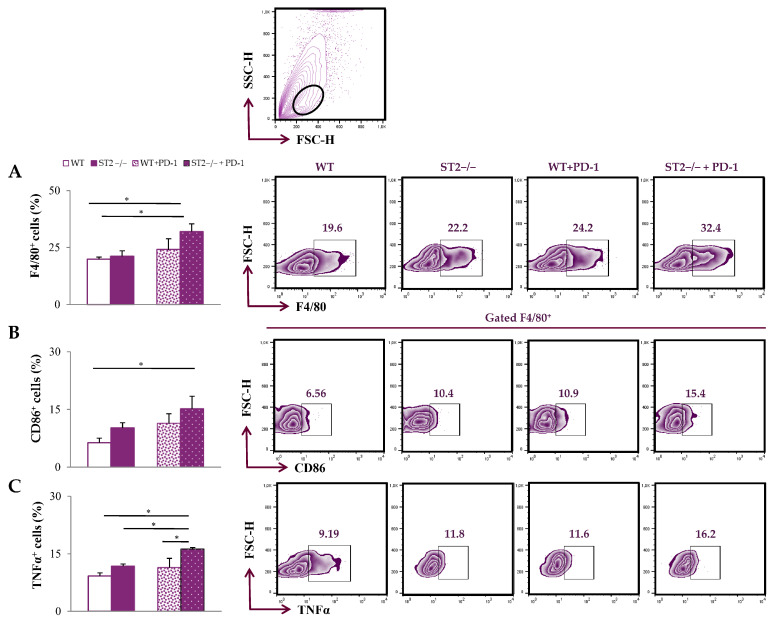
Dual blockade of the PD-1/PD-L and IL-33/ST2 axes enhances the accumulation of M1 macrophages within the tumor microenvironment. The graphs and representative plots represent the percentage of F4/80^+^ cells (**A**), CD86^+^F4/80^+^ cells (**B**) and expression of TNFα in F4/80^+^ cells (**C**) derived from primary breast tumor of WT (BALB/C), ST2^−/−^ (BALB/C ST2 knockout), anti-PD-1-treated WT (WT + PD-1), and anti-PD-1-treated ST2^−/−^ (ST2^−/−^ + PD-1) mice. The upper panel contains a fact plot with a gating strategy for mononuclear cells (leukocytes). Data are presented as average ± SEM (*n* = 7 mice per group). Statistical significance was tested via the Kruskal–Wallis test and post hoc Mann–Whitney Rank Sum test. * *p* < 0.05.

**Figure 3 ijms-26-09600-f003:**
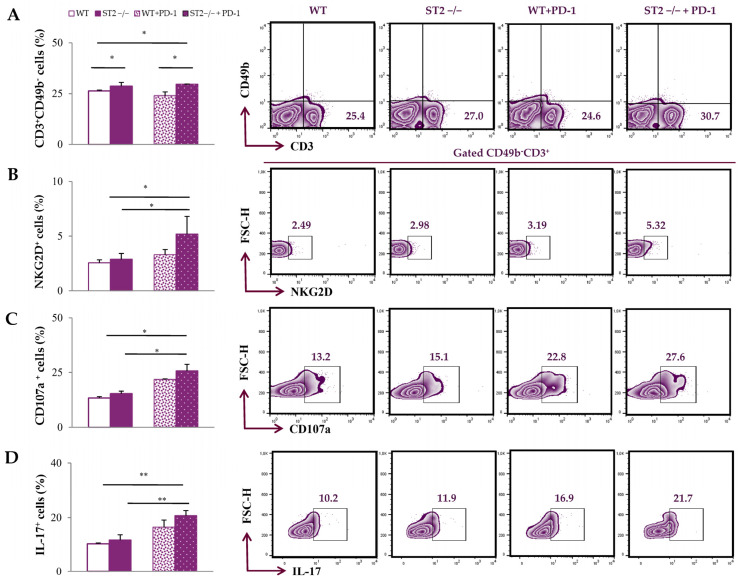
Anti-PD-1 treatment increases the percentage of T cells and the expression of activation molecules in the spleens of ST2^−/−^ mice. The graphs and representative plots represent the percentage of CD3^+^CD49b^−^ cells (**A**), NKG2D^+^CD3^+^CD49b^−^ cells (**B**), CD107a^+^ CD3^+^CD49b^−^ cells (**C**), and IL-17^+^ CD3^+^CD49b^−^ cells (**D**) derived from spleens of WT (BALB/C), ST2^−/−^ (BALB/C ST2 knockout), anti-PD-1-treated WT (WT + PD-1), and anti-PD-1-treated ST2^−/−^ (ST2^−/−^ + PD-1) mice. Data are presented as average ± SEM (*n* = 7 mice per group). Statistical significance was tested by Kruskal–Wallis test and post hoc Mann–Whitney Rank Sum test.* *p* < 0.05; ***p* < 0.01.

**Figure 4 ijms-26-09600-f004:**
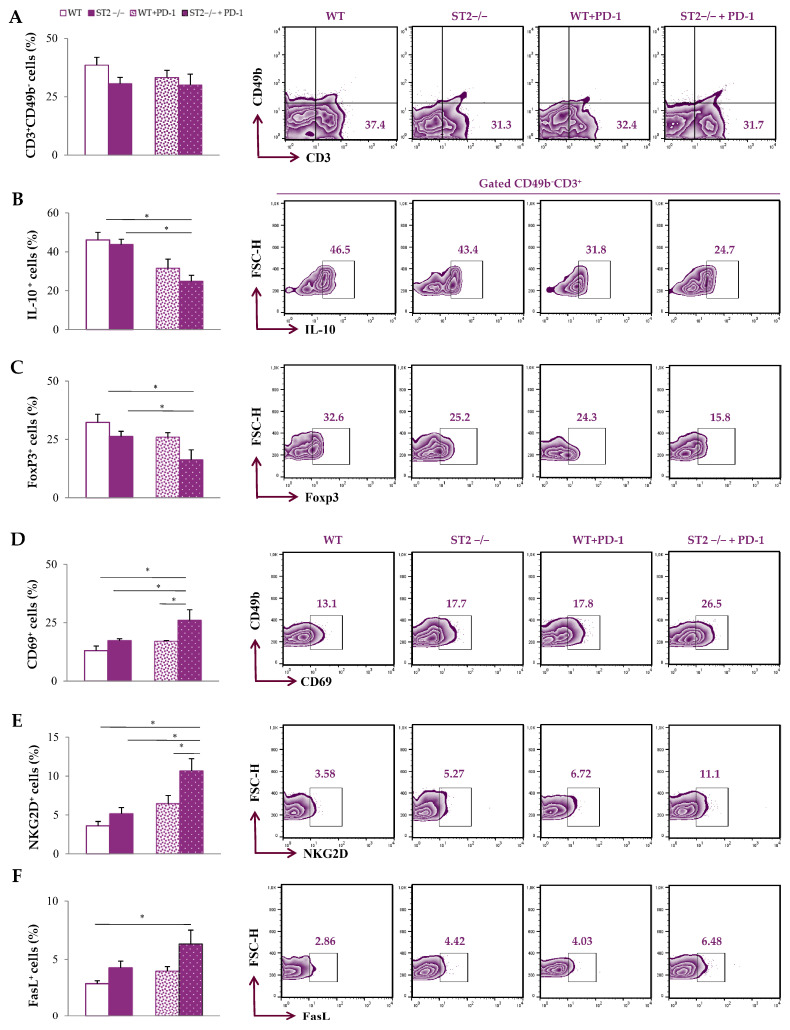
Pro-tumoricidal phenotypes of T cells in the tumor microenvironment of anti-PD-1-treated ST2^−/−^ mice. The graphs and representative plots represent the percentage of CD3^+^CD49b^−^ cells (**A**), IL-10^+^CD3^+^CD49b^−^ cells (**B**), FoxP3^+^CD3^+^CD49b^−^ cells (**C**), CD69^+^CD3^+^CD49b^−^ cells (**D**), NKG2D^+^CD49b^−^ cells (**E**), and FasL^+^CD3^+^CD49b^−^ cells (**F**) derived from primary tumors of WT (BALB/C), ST2^−/−^ (BALB/C ST2 knockout), anti-PD-1-treated WT (WT + PD-1), and anti-PD-1-treated ST2^−/−^ (ST2^−/−^ + PD-1) mice. Data are presented as average ± SEM (*n* = 7 mice per group). Statistical significance was tested by Kruskal–Wallis test and post hoc Mann–Whitney Rank Sum test. * *p* < 0.05.

**Figure 5 ijms-26-09600-f005:**
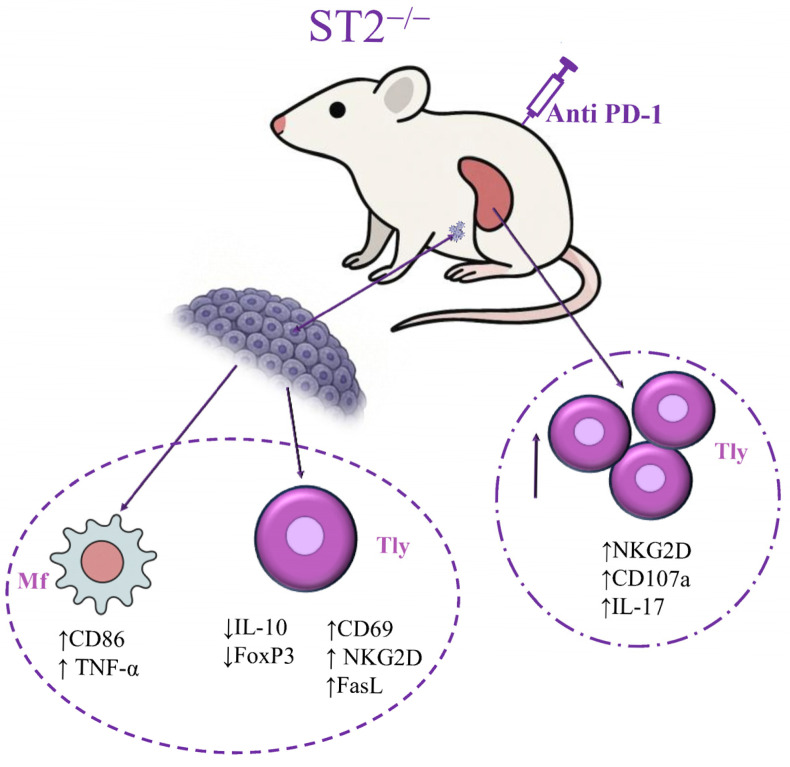
Dual blockade stimulates acquired anti-tumor immunity by boosting T cells’ activity and macrophage polarization towards the M1 phenotype, which in turn can also help to stimulate overall T cell activity, leading to more efficient anti-tumor response.

## Data Availability

The raw data supporting the conclusions of this article will be made available by the authors on request.

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
