# Peer review of "Co-Targeting PD-1 and IL-33/ST2 Pathways for Enhanced Acquired Anti-Tumor Immunity in Breast Cancer"

_ijms, 2025, doi:10.3390/ijms26199600_

Round 1
Reviewer 1 Report
Comments and Suggestions for Authors
Overall Impression: Authors presented interesting and potentially significant findings on the combined effects of anti-PD-1 therapy and ST2 deficiency. The data suggests that this dual blockade leads to a more robust anti-tumor immune response by modulating both macrophage and T cell populations. The experiments are well-designed to address the hypotheses.
Minor Comments:
- The introduction to the results section is a bit abrupt. It would be beneficial to have couple of introductory lines to show why the authors chose to investigate the dual blockade of the PD-1/PD-L and IL-33/ST2 axes.
- A supplementary figure describing how F4/80+, CD3+CD49b- populations were gated would add clarity to the results.
- English need to be checked for spelling and grammar
- Line 126: "anlyzes" should be "analyzed."
- Line 127: "CD3+CD49-" should be "CD3+CD49b-".
- Line 133: "pergentage" should be "percentage."
- Line 134: "anti-PD-1tretment" should be "anti-PD-1 treatment."
- Line 149: "ocuring" should be "occurring."
- Line 150: "controling" should be "controlling."
- Line 151: "chemoresitance" should be "chemoresistance."
- Line 168: "imortant" should be "important."
- Line 178: "imlicate" should be "implicate."
- Line 186: "esspecially" should be "especially."
- Line 208: is it "IL-33/PD1 and PDL/PD1" or "IL-33/ST2 and PDL/PD1"
- Line 241: "efficent" should be "efficient."
None
Author Response
The introduction to the results section is a bit abrupt. It would be beneficial to have couple of introductory lines to show why the authors chose to investigate the dual blockade of the PD-1/PD-L and IL-33/ST2 axes.
Response: We thank the reviewer for this valuable suggestion. Additional explanation for the investigation of dual blockage was added in the Manuscript.
A supplementary figure describing how F4/80+, CD3+CD49b- populations were gated would add clarity to the results.
Response: We have now incorporated a supplementary figure into the manuscript, which illustrates the gating strategy for F4/80+ and CD3+CD49b− populations in spleen as well as in primary tumor.
English need to be checked for spelling and grammar
Line 126: "anlyzes" should be "analyzed."
Line 127: "CD3+CD49-" should be "CD3+CD49b-".
Line 133: "pergentage" should be "percentage."
Line 134: "anti-PD-1tretment" should be "anti-PD-1 treatment."
Line 149: "ocuring" should be "occurring."
Line 150: "controling" should be "controlling."
Line 151: "chemoresitance" should be "chemoresistance."
Line 168: "imortant" should be "important."
Line 178: "imlicate" should be "implicate."
Line 186: "esspecially" should be "especially."
Line 208: is it "IL-33/PD1 and PDL/PD1" or "IL-33/ST2 and PDL/PD1"
Line 241: "efficent" should be "efficient."
Response: We would like to thank the reviewer for pointing out our mistakes. We have now checked our English for grammar and spelling, and also have spell-corrected all of the mistakes listed above.
Reviewer 2 Report
Comments and Suggestions for Authors
This is a review of the manuscript by Jovanovic et al., titled “Co-targeting PD-1 and IL-33/ST2 pathways for enhanced acquired anti-tumor immunity in breast cancer.”
The dose concentration of anti-PD-1 is concerning. As written, it appears that all mice received the same dose and it was not calculated based on the weights of the animals. Instead, it appears that a uniform dose was administered. This is highly irregular as most treatment concentrations are determined by animal weight.
There is no background given as to why Balb/c ST2 knockout mice were used.
Only flow cytometer data is presented yet one of the conclusions drawn is that inhibiting PD-1 and ST2 “alters tumor microenvironment and 243 stimulates overall anti-tumor immune response.” However ,no data is presented indicating that tumor sizes change or that the cancer cells have any change in phenotype, proliferation, or cell death. They demonstrate that the immune cell populations have changed but not that any anti-tumor effects have.
Comments on the Quality of English LanguageThe English requires moderate editing for grammar and spelling.
Author Response
The dose concentration of anti-PD-1 is concerning. As written, it appears that all mice received the same dose and it was not calculated based on the weights of the animals. Instead, it appears that a uniform dose was administered. This is highly irregular as most treatment concentrations are determined by animal weight.
Response: We thank the reviewer for raising this point. We initially have decided on the given dose based on the work of Qin L et al, which also investigates both of these axes in a context of highly immunogenic tumor, where is PD1 blockage combined with exogenous stimulation of IL33/ST2 axis (Qin, L., Dominguez, D., Chen, S., Fan, J., Long, A., Zhang, M. et al. Exogenous IL-33 overcomes T cell tolerance in murine acute myeloid leukemia. Oncotarget. 2016;7(38):61069‐61080. doi:10.18632/oncotarget.11179). In addition, the given dose was used in our previously published study (Jovanovic MZ, Geller DA, Gajovic NM, Jurisevic MM, Arsenijevic NN, Jovanovic MM, Supic GM, Vojvodic DV, Jovanovic IP. Dual blockage of PD-L/PD-1 and IL33/ST2 axes slows tumor growth and improves antitumor immunity by boosting NK cells. Life Sci. 2022 Jan 15;289:120214. doi: 10.1016/j.lfs.2021.120214). This approach was selected to ensure comparability with previously published studies, and because body-weight differences among the mice in our experiments were minimal (within the 19–21 g range).
There is no background given as to why Balb/c ST2 knockout mice were used.
Response: Thank you for your comment. BALB/c ST2 knockout mice were used in our study because the deletion of the ST2 gene effectively blocks the IL-33/ST2 signaling pathway. This genetic model allowed us to investigate the contribution of IL-33 signaling to tumor progression and antitumor immunity. The explanation is now also provided in Materials and methods section (section 4.1).
Only flow cytometer data is presented yet one of the conclusions drawn is that inhibiting PD-1 and ST2 “alters tumor microenvironment and 243 stimulates overall anti-tumor immune response.” However,no data is presented indicating that tumor sizes change or that the cancer cells have any change in phenotype, proliferation, or cell death. They demonstrate that the immune cell populations have changed but not that any anti-tumor effects have.
Response: Thank you for your comment. We have investigated earlier the potent effects of dual blockage of IL33/ST2 and PDL/PD1 on tumor appearance and growth, as illustrated in a study by Jovanovic M et al. (Jovanovic MZ, Geller DA, Gajovic NM, Jurisevic MM, Arsenijevic NN, Jovanovic MM, Supic GM, Vojvodic DV, Jovanovic IP. Dual blockage of PD-L/PD-1 and IL33/ST2 axes slows tumor growth and improves antitumor immunity by boosting NK cells. Life Sci. 2022 Jan 15;289:120214. doi: 10.1016/j.lfs.2021.120214). In that study, we illustrated the effects of dual blockade on innate antitumor immunity and tumor progression. In contrast, the current manuscript is designed to extend these findings by focusing on acquired antitumor immunity, specifically the phenotype changes in T cells and on macrophages. Nevetheless, the explanation given above is now incorporated in disscusion section.
Round 2
Reviewer 2 Report
Comments and Suggestions for Authors
The issues raised initially have been adequately addressed, except for the improvement of the English. There are still multiple instances of incorrect grammar that should be addressed before acceptance.
Comments on the Quality of English LanguageThere are still multiple instances of incorrect grammar that should be addressed before acceptance.